# A Reduced Starch Level in Plants at Early Stages of Infection by Viruses Can Be Considered a Broad-Range Indicator of Virus Presence

**DOI:** 10.3390/v14061176

**Published:** 2022-05-28

**Authors:** Wanying Zhao, Li Wang, Meizi Liu, Dong Zhang, Ida Bagus Andika, Ying Zhu, Liying Sun

**Affiliations:** 1State Key Laboratory of Crop Stress Biology for Arid Areas, College of Plant Protection, Northwest A&F University, Xianyang 712100, China; zwy01@nwafu.edu.cn (W.Z.); wangli_nwafu@163.com (L.W.); 2State Key Laboratory for Managing Biotic and Chemical Threats to the Quality and Safety of Agro-Products, Institute of Virology and Biotechnology, Zhejiang Academy of Agricultural Sciences, Hangzhou 310024, China; 3Yangling Sub-Center of National Center for Apple Improvement, College of Horticulture, Northwest A&F University, Yangling, Xianyang 712100, China; z13102981768@163.com (M.L.); afant@nwafu.edu.cn (D.Z.); 4College of Plant Health and Medicine, Qingdao Agricultural University, Qingdao 266109, China; idabagusyf@yahoo.com

**Keywords:** plant virus, diagnosis of plant viral disease, detection method, starch accumulation, iodine staining

## Abstract

**Simple Summary:**

In this report, we describe a potential broad-range indicator of virus presence in plants. This method is based on the observation that starch content in leaves is generally reduced upon virus infection at the early stages of infection. This method was also used to effectively detect the infection of three latent viruses in propagative apple materials. Hence, the plant with the low virus accumulation, latent infection or at early stage of infection can be indicated by assessing starch accumulation levels. The genes encoding key enzymes involved in starch biosynthesis are transcriptionally downregulated by virus infection, suggesting that plant virus infection generally affects cellular metabolic pathways.

**Abstract:**

The diagnosis of virus infection can facilitate the effective control of plant viral diseases. To date, serological and molecular methods for the detection of virus infection have been widely used, but these methods have disadvantages if applied for broad-range and large-scale detection. Here, we investigated the effect of infection of several different plant RNA and DNA viruses such as cucumber mosaic virus (CMV), tobacco mosaic virus (TMV), potato virus X (PVX), potato virus Y (PVY) and apple geminivirus on starch content in leaves of *Nicotiana benthamiana*. Analysis showed that virus infection at an early stage was generally associated with a reduction in starch accumulation. Notably, a reduction in starch accumulation was readily apparent even with a very low virus accumulation detected by RT-PCR. Furthermore, we also observed that the infection of three latent viruses in propagative apple materials was associated with a reduction in starch accumulation levels. Analysis of transcriptional expression showed that some genes encoding enzymes involved in starch biosynthesis were downregulated at the early stage of CMV, TMV, PVX and PVY infections, suggesting that virus infection interferes with starch biosynthesis in plants. Our findings suggest that assessing starch accumulation levels potentially serve as a broad-range indicator for the presence of virus infection.

## 1. Introduction

As the second most important plant disease agents, viruses pose a major threat to agricultural production by causing losses of yield and reducing product quality, which lead to economic losses more than several billions dollars worldwide every year [1]. Unlike fungi and bacteria, which can be controlled using chemical treatments, no effective method is available to cure plants upon virus infection. Therefore, the detection of virus infection is essential for control strategies aimed at limiting viral emergence and spread in the agricultural production system. Varieties of plant virus detection methods have been developed. In general, virus particles can be observed using an electron microscope, while viral proteins can be detected by immunological methods, including enzyme-linked immunosorbent assay, immunofluorescence, and electro-blot immunoassay. Viral nucleic acids can be detected using molecular methods, such as northern blotting, reverse transcription-polymerase chain reaction (RT-PCR), and loop-mediated isothermal amplification [2]. Observation of the viral structure can facilitate the diagnosis of viruses. However, compared to other plant pathogens such as fungi and bacteria, which can be easily observed with a light microscope, viruses can only be visualized by electron microscopy. This method requires an expert in using an electron microscope. Therefore, serological and molecular methods are more widely used for the detection of virus infections.

Up to date, technologies related to vegetative or clonal propagation have been developed for many important food crops, including herbs, shrubs, trees and vines. However, vegetative propagates, such as bulbs, corms, tubers, cuttings, and buds, are often contaminated with viruses that then spread throughout the fields [3]. Hence, the early detection of viruses in propagative materials and the utilization of virus-free materials are crucial to avoid viral infections. Although the abovementioned techniques have been widely used, they are less suitable for high-throughput testing and the simultaneous detection of numerous viruses. Thus, faster and more efficient detection techniques are necessary for the large-scale detection of plant viruses, particularly for the production of virus-free vegetative propagates.

Viruses are obligate parasites that depend on host cell components for their multiplication. To hijack the cellular machinery, viruses develop numerous strategies that modify cellular metabolism [4,5]. Redirection of the carbon supply used in cellular metabolic pathways is crucial for viral replication [6]. Virus infection leads to a wide range of biochemical and physiological changes in photosynthesis, CO_2_ assimilation, and starch accumulation [7]. The common symptom induced by virus infection is leaf chlorosis, which is often associated with disturbances of chloroplast activities. Previously, the symptoms in tobacco plants caused by cucumber mosaic virus (CMV) were found to be associated with reduced starch accumulation level. Using iodine staining of leaves, it was observed that the starch accumulation was increased in CMV inoculated leaves but was decreased in systemically infected leaves [8]. Electron microscope observation showed that turnip yellow mosaic virus (TYMV) caused a reduction in the number of starch grains in chloroplasts of the cells of the apical dome of Chinese cabbage seedling [9]. In addition, other previous reports showed that virus infection resulted in changes in starch accumulation in host plants [10,11,12]. 

Herein, we found that starch accumulation levels in systemically infected leaves was generally suppressed upon infection with various plant viruses at the early stages of viral infection. Based on this observation, we developed a diagnostic method involving the assessment of the starch content in newly growing leaves that potentially indicate a broad range of plant viruses in virus-contaminated propagative materials in a relatively sensitive and easy manner. 

## 2. Materials and Methods

### 2.1. Plant and Virus Materials

*Nicotiana. benthamiana* plants were grown in soil and kept in plant chambers with 12 h of daylight at 25 °C. The shoots of apple tissue culture (*Malus domestica* Borkh) were obtained from the Yangling Sub-center of the National Center of Apple Improvement, Shaanxi, China. The tissue culture plants were prepared and maintained on Murashige and Skoog (MS) agar medium containing (0.2 mg/L 6-Benzylaminopurine and 0.2 mg/L indole-3-acetic acid) and kept at 23 °C, with 16/8 h (light/dark) photoperiod, as described previously [13,14].

The plant RNA viruses, CMV isolate Fny (accession No. MG025947, MG025948 and MG025949 for RNA1, RNA2 and RNA3, respectively) was provided by Xianbin Wang (China Agricultural University). Tobacco mosaic virus (TMV), potato virus X (PVX) and potato virus Y (PVY) are laboratory collections isolated from potato plants (Solanum tu-berosum), which were obtained from Yulin City, Shaanxi Province, China. The plant DNA virus, apple geminivirus (AGV), was isolated from an apple plant in Yangling City, Shaanxi Province. Sequence analysis of coat protein gene showed that these TMV, PVX, PVY and AGV isolates have 100%, 98%, 99% and 100% identities to TMV isolate accession No. AF165190, PVX isolate accession No. NC_011620.1, PVY isolate accession No. JF927760.1 and AGV isolate accession No. KM386645, respectively.

### 2.2. Virus Inoculation and Agro-Infiltration 

For mechanical inoculation of RNA viruses, leaves of *N. benthamiana* at the six-leaf stage were dusted with carborundum powder and rubbed with purified CMV, TMV, PVX, and PVY particles diluted in 20 mM sodium phosphate buffer pH 7.0. The resulting *N. benthamiana* leaves infected with CMV, TMV, PVX and PVY were stored at −80 °C and used as the virus inoculum. The AGV (Yangling isolate) was inoculated using an infectious DNA clone prepared as previously described [15] and transformed into *Agrobacterium tumefaciens* (strain GV3101). Agro-infiltration was performed as described previously [16].

### 2.3. Sample Preparation for Starch Purification and Nucleic Acid Extraction

The lower leaves of *N. benthamiana* plants at the six-leaf stage were inoculated with viruses or mock-inoculated and the upper leaves were collected from the plants with or without virus inoculation at 3, 6, 12, and 24 h post-inoculation (hpi) or 3, 7, 14 days post-inoculation (dpi). Each half portion of leaf was used for starch purification and RNA extraction.

### 2.4. RNA Extraction and RT-PCR

Total RNAs were extracted from two leaves obtained from at least three plants, using Trizol (Invitrogen, Carlsbad, CA, USA) according to the manufacturer’s protocol. The 1 µg total RNA was used as the template for reverse transcription by using ReverTra Ace reverse transcriptase (Toyobo, Japan). PCR was carried out using 2xTaq Plus MasterMix (Dye) (Cowin Biotech Co., Ltd., Beijing, China). PCR amplification was carried out as follows. The initial denaturation step at 94 °C for 2 min, then cycle of denaturation at 94 °C for 20 s, annealing at 58 °C for 20 s, extension at 72 °C for 1 min and final extension at 72 °C for 5 min. PCR amplification cycles for AGV, ASGV, ASPV and ACLSV are 35 cycles and for Nb18S, CMV, TMV, PVY, PVX, and MdEF-1α are 32 cycles. The sequences of the primers used for the virus detection and quantitative gene expression analysis are listed in Appendix A. For quantitative RT-PCR, 18S rRNA of *N. benthamiana* (TC23401) and EF-1α (MdEF-1α) (NC_041799) of *Malus domestica* Borkh (NC_041799) were used as internal controls [13,17]. Each cDNA was diluted 10-fold and used as a template. PCR reactions were conducted with the GoTaq^®^ Green Master Mix kit (Promega, Madison, WI, USA) on a CFX96TM Real-Time PCR Detection System apparatus (Bio-Rad, Hercules, CA, USA). 

### 2.5. DNA Preparation and PCR

The DNA was extracted from plant leaves using a cetyltrimethylammonium bromide (CTAB) buffer, as described previously [18]. PCR reactions were performed with DNA polymerase (TAKARA, Beijing, China). 

### 2.6. Calibration of Starch Standard Curve

The calibration of the starch standard curve was plotted for mixtures of amylose and amylopectin of the tapioca flour (Zhengzhou Niuji Condiment Co., Ltd., Zhengzhou, China) solved in Milli-Q water with concentrations 0, 0.1, 0.2, 0.4, 0.6, 0.8, 1.0 mg/mL. Then, 500 μL of each starch solution was mixed with 500 μL of 80% calcium nitrate medium and 200 μL of 0.5% iodine solution and incubated for 15 min. The starch was pelleted and washed twice with a 500 μL solution containing 5% Ca(NO_3_)_2_ and 0.01% iodine and then dissolved in 1 mL of 0.1 mM NaOH. To the starch solution, 15 μL of 0.5% iodine solution and 200 μL of 1 mM HCl were added, followed by water to make a final volume of 2 mL. All samples were scanned using UV-visible spectrophotometric over the wavelength range 270–900 nm. The absorption maxima of the iodine staining mixtures was determined at 560 nm. The absorbance of the starch solution was then read in a quartz cell at 560 nm using a UNICO UV/visible spectrophotometer (UV-2600A; www.unicosh.com.cn, (accessed on 24 May 2016)). The calibration graph of absorbance at 560 nm against starch solution (0–1 mg/mL) was carried out. The correlation coefficient was above 0.999.

### 2.7. Starch Purification and Quantification

For starch quantification, plant samples (infected or non-infected) were generated using the procedure described above. The starch was extracted from fresh leaves as previously described [19], with modification. The procedure for the quantification of starch content in leaves consists of 10 steps (Appendix A). In detail, Step 1: 10–30 mg leaf tissues are homogenized in 500 µL of 80% Ca(NO_3_)_2_. Step 2: The homogenate is transferred to a 2-mL Eppendorf tube and heated in a boiling water bath for 5 min. Step 3: 650 µL Milli-Q water is added and the sample is centrifuged at 3000 rpm for 2–3 min. The resulting supernatant is transferred to a new 2-mL Eppendorf tube and the pellet is diluted with 1 mL Milli-Q water and then centrifuged at 3000 rpm for 2–3 min. The supernatant is transferred to the same tube. Step 4: Milli-Q water is added up to a volume of 2 mL and then centrifuged at 3000 rpm for 2–3 min. Step 5: 200 µL of 0.5% iodine solution (0.5 g iodide and 10 g potassium iodide/L water) is added to 500 µL of the sample solution and incubated for 15 min. Step 6: The sample is centrifuged at 3000 rpm for 5 min, and then the supernatant is removed. Step 7: The pellet is resuspended in 500 µL solution containing 0.01% iodine and 5% Ca(NO_3_)_2_ and centrifuged at 3000 rpm for 3 min. This step is performed twice. Step 8: The pellet is dissolved in 1 mL of 0.1 mM NaOH by heating in a boiling water bath for 5 min. Step 9: 15 µL of 0.5% iodine solution and 200 µL of 1 mM HCl are added, along with Milli-Q water up to a final volume of 2 mL. Step 10: 100 µL aliquot of the solution is transferred to a 1 cm pathlength quartz cell and the absorbance is read at 560 nm using a UNICO UV/visible spectrophotometer.

## 3. Results and Discussion

Starch contains two types of polysaccharides: amylose, a linear chain of glucose, and amylopectin, a highly branched chain of glucose. A widely used method for the quantification of starch is a colorimetric assay in which amylose and amylopectin bind to iodine and generate a blue-purple color that is measured spectrophotometrically at wavelengths ranging from 540 nm to 620 nm. This method is relatively simple, rapid, and inexpensive. It can be applied for most plant tissues. Hence, in this study, the starch content in the leaves was measured by spectrophotometry and calculated using a calibration curve [20]. The tapioca flour was used as the amylose and amylopectin standard for starch quanti-fication. The standard curve was plotted according to the series of tapioca flour solutions (Appendix A), and the absorbance of the iodine stain was analyzed by using a spectrophotometer and measured at 560 nm for obtaining the maxima absorbance. From the absorbance curve, the linear regression equation Y (starch concentration, mg/mL) = 0.412X (absorbance) − 0.06 (correlation coefficient, r = 0.9991) was obtained and used to calculate the starch content (Appendix A).

Virus infection commonly interferes with the structure and function of chloroplast and thus induces typical photosynthesis-related symptoms, such as chlorosis and mosaic [21]. As starch is synthesized in the chloroplast, virus infection may influence starch accumulation in the plant. To investigate the general effects of virus infection on starch accumulation, different types of plant viruses, including the RNA viruses, CMV, TMV, PVX, and PVY and a latent DNA virus, AGV, were inoculated into *N. benthamiana* plants. Based on the viral symptom development, in this study, virus infection was divided into three stages; namely, early stage (2–3 days), symptom-emerging stage (6–7 days), and developed-symptom stage (≥14 days; Figure 1A). The accumulation of starch during photosynthesis and its linear degradation at night is crucial for stable plant growth [22]. Therefore, all plants were grown in plant chambers with a 12 h light-dark cycle at room temperature, and the leaf samples were collected 2–3 h before the dark period. After inoculation, virus infection was confirmed by RT-PCR detection (Figure 1B, Appendix A). Newly emerged upper leaves were sampled for analysis. As shown in Figure 1C, at the early and symptom-emerging stages, the starch content in the plants infected with the five viruses was significantly lower compared to that of mock-inoculated plants. At the symptom-developed stage, the starch content was significantly lower in plants infected with CMV, PVX, and AGV than in mock-inoculated plants, but higher in plants infected with TMV and PVY than in mock-infected plants. The overall reduction in starch accumulation caused by virus infection at the initial and symptom-emerging stages suggests that virus infection generally affects photosynthesis at the early stages of infection. However, at the later stage of viral infection, different viruses seem to have varying effects on starch content. Thus, reduction in starch content could be a general indicator of virus infection at the early stage. The distinct effects on starch accumulation between different infection stages may reflect the complex and dynamic interactions between viruses and host phytohormonal, nutritional, and stress signaling pathways [9,23,24]. Thus, the distinct regulation of leaf starch metabolism by virus infection is the consequence of the specific interactions between virus and host.

Time-course analysis showed that PVX was detected by RT-PCR in the upper systemic leaves of PVX-inoculated plants as early as 6 h post inoculation (hpi; Figure 2A, Appendix A). Accordingly, a reduction in starch content in the systemic leaves of PVX-inoculated plants was observed from 6 hpi onward (Figure 2B). This result is similar with the reduction in starch grain numbers in the apical dome cells of Chinese cabbage seedlings at 6–8 h after inoculation with TYMV [9]. When the PVX inoculum was subjected to 10-fold serial dilution and inoculated into *N. benthamiana* plants, a significant reduction in starch content in the upper leaves was observed only up to the 10-fold dilution of the inoculum (Figure 2C), which was in line with the clear detection of PVX by RT-PCR at this dilution but not at higher dilutions (Figure 2D). Together, these results show that in the case of PVX, the sensitivity of starch content assessment to indicate virus infection is comparable to that of the RT-PCR method. Therefore, this technique is potential to be developed as a novel diagnostic for early stage plant virus infections. 

To test the suitability of this method for virus detection in propagative plant materials, 5- to 8-week-old virus-free and virus-infected shoots of apple tissue culture were obtained (Figure 3A). Three latent viruses were previously found to be widely distributed in apple plants in the Yangling Sub-center, including apple stem pitting virus (ASPV), apple stem grooving virus (ASGV), and apple chlorotic leaf spot virus (ACLSV) [25]. The resulting tissue-culture-derived apple shoots were simultaneously tested by RT-PCR using primers (Appendix A) specific for the gene of the viral coat protein (Figure 3B) and examined for their starch content (Figure 3C). The results showed that starch accumulation levels were consistently lower in shoots infected with either single or multiple viruses than that in virus-free shoots (Figure 3B,C). This result indicates that the latent virus infection in propagative materials can be detected by monitoring starch accumulation levels. 

Starch metabolism in plants involves a series of enzyme activities, which are functionally classified into synthetic and degradative reactions [26,27]. Starch synthesis starts with the substrate of glucose-1-phosphate (GlclP) that is generated directly from the photosynthatetical Calvin–Benson cycle through the conversion of fructose-6-phosphate (Fru6p) to glucose-6-phosphate (Glu6P) during the daytime. The adenosine 5′-diphosphate glucose pyrophosphorylase (AGPase) catalyzes the conversion of GlclP and ATP to ADPglucose (ADP-Glc) and pyrophosphate (PPi) (Figure 4A). The ADP-Glc is then converted to amylose and amylopectin by granule-bound starch synthase (GBSS), Starch synthase (SS), and starch branching enzyme (BE). At night, the starch is degraded. It is suggested that the starch degradation consists of two phases: initiation of degradation, and digestion of amylopectin and amylose. The results of starch degradation components such as maltose, glucose and maltotriose are transported into cytoplasm (Figure 4A) and play an important role in extracellular and intracellular activity. In the initiation of starch degradation, the starch dikinases (GWD/PWD) play a crucial role. To determine whether the activities of starch metabolism are affected by virus infection, the expression of several genes encoding enzymes involved in starch biosynthesis and degradation, including AGPase, GBSS, beta-amylase, isoamylasarce 3, and glucan, starch dikinase (GWD) in *N. benthamiana* following PVX, CMV, TMV, and PVY infection were analyzed by quantitative RT-PCR. PVX and CMV infections were associated with an overall reduction in transcript expressions of those genes at 3 dpi and, to a greater extent, at 14 dpi (Figure 4B,C). In contrast, TMV infection upregulated the transcript expressions of some AGPase genes (AGPase L1 and L2) at both 3 and 14 dpi, while it down regulated GBSS gene expression at 3 dpi but not at 14 dpi (Figure 4D). This observation likely explains why TMV infection is associated with the reduction in starch accumulation levels at 3 dpi and the increase in starch levels at 14 dpi (Figure 1C). In PVY infected plants, which also showed a reduction in starch accumulation levels at 3 dpi but an increase in starch levels at 14 dpi (Figure 1C), transcript expressions of some AGPase genes (AGPase L1 and S) and GBSS genes were reduced at 3 dpi, but none of the genes was upregulated at 14 dpi (Figure 4E). Notably, following PVY infection, the expression of the GWD gene, which is involved in starch degradation, was reduced at 3 dpi and to a much larger extent at 14 dpi (Figure 4E). Thus, the increase in starch accumulation at 14 dpi during PVY infection may be partially due to the inhibition of the starch degradation process. Thus, it appears that the general reduction in starch accumulation at the early stages of virus infection is caused by the suppression of transcript expressions of particular genes involved in starch biosynthesis; however, at later stages of infection, different viruses have varying effects on gene expressions. These results provide insight into the mechanism by which virus infection interferes with starch biosynthesis in plants. 

## 4. Conclusions

This study demonstrates that plant viruses generally suppress starch accumulation in leaves at the early stage of infection. Based on this observation, it is suggested that assessing starch accumulation levels potentially serves as a broad-range detection method to diagnose plant virus infection, particularly at the early stage of infection. In the case of PVX, the sensitivity of this method is comparable to that of PCR detection. Moreover, this method was successfully applied to detect latent viruses in propagative apple tissue culture materials.

## Figures and Tables

**Figure 1 viruses-14-01176-f001:**
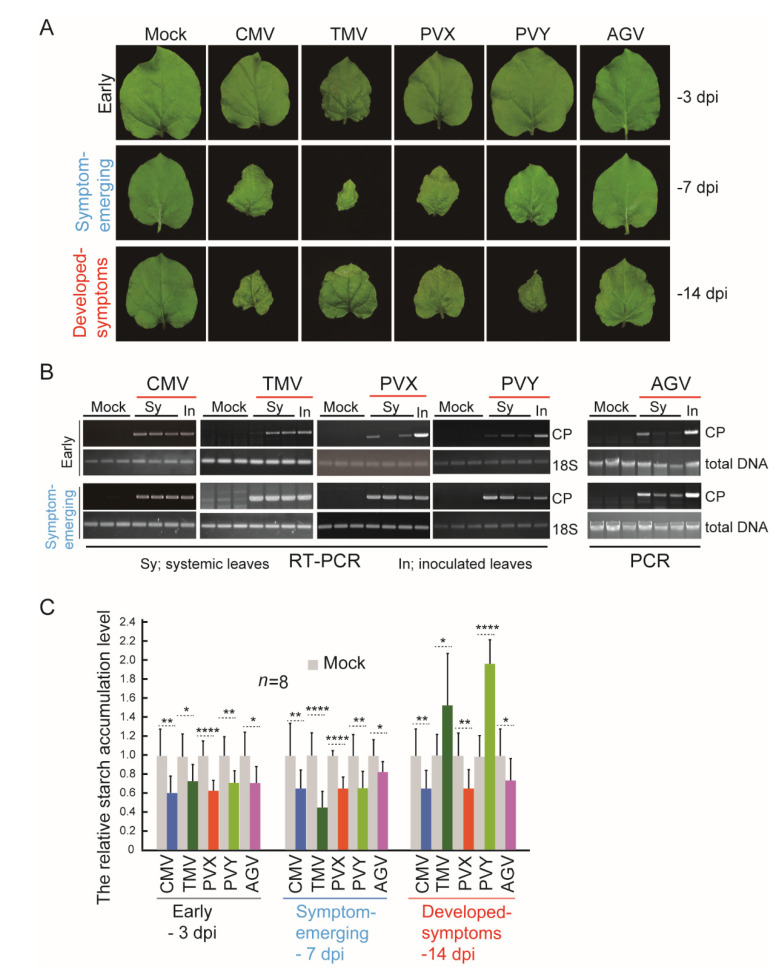
Effects of plant virus infection on starch accumulation levels. (**A**) Virus symptom phenotypes on the *N. benthamiana* leaves collected for starch content analysis and RT-PCR. (**B**) RT-PCR detection of viruses in the plants described in (**A**) using primer sets specific for the viral coat protein gene. Amplification of *N. benthamiana* 18S was used as a control standard. PCR products are separated on agarose gels via electrophoresis and stained with ethidium bromide (EtBr) (**C**) The relative starch accumulation in the leaves described in (**A**). Data are presented as mean ± SD. The mock sample data value was set to 1.0. *, ** and **** indicate a significant difference at *p* < 0.05, 0.01, 0.001, or 0.0001, respectively (Student’s *t*-test).

**Figure 2 viruses-14-01176-f002:**
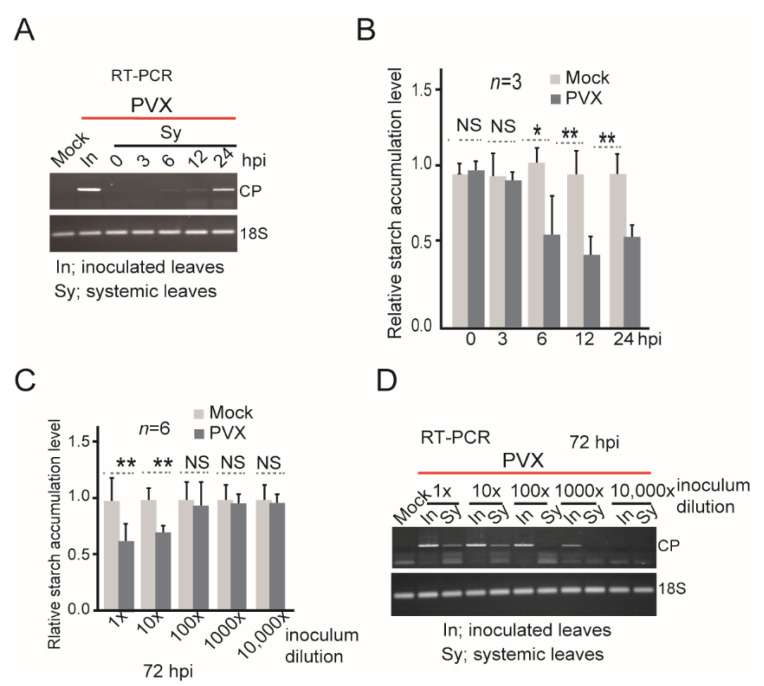
The sensitivity of virus detection method by assessing starch accumulation levels. (**A**) Time-course RT-PCR detection of PVX in upper leaves at 0, 3, 6, 12, and 24 h post inoculation (hpi). (**B**) Relative starch accumulation in the leaves described in (**A**). Data are presented as mean ± SD. The mock sample data value was set to 1.0. (**C**) Relative starch accumulation in the leaves of plants inoculated with 10-fold serially diluted PVX inoculum. Data are presented as mean ± SD. The mock sample data value was set to 1.0. (**D**) RT-PCR detection of PVX in the leaves of plants described in (**C**). “*”, and “**”, indicate a significant difference at *p* < 0.05, and 0.01, respectively. “NS” indicate no significant difference (Student’s *t*-test).

**Figure 3 viruses-14-01176-f003:**
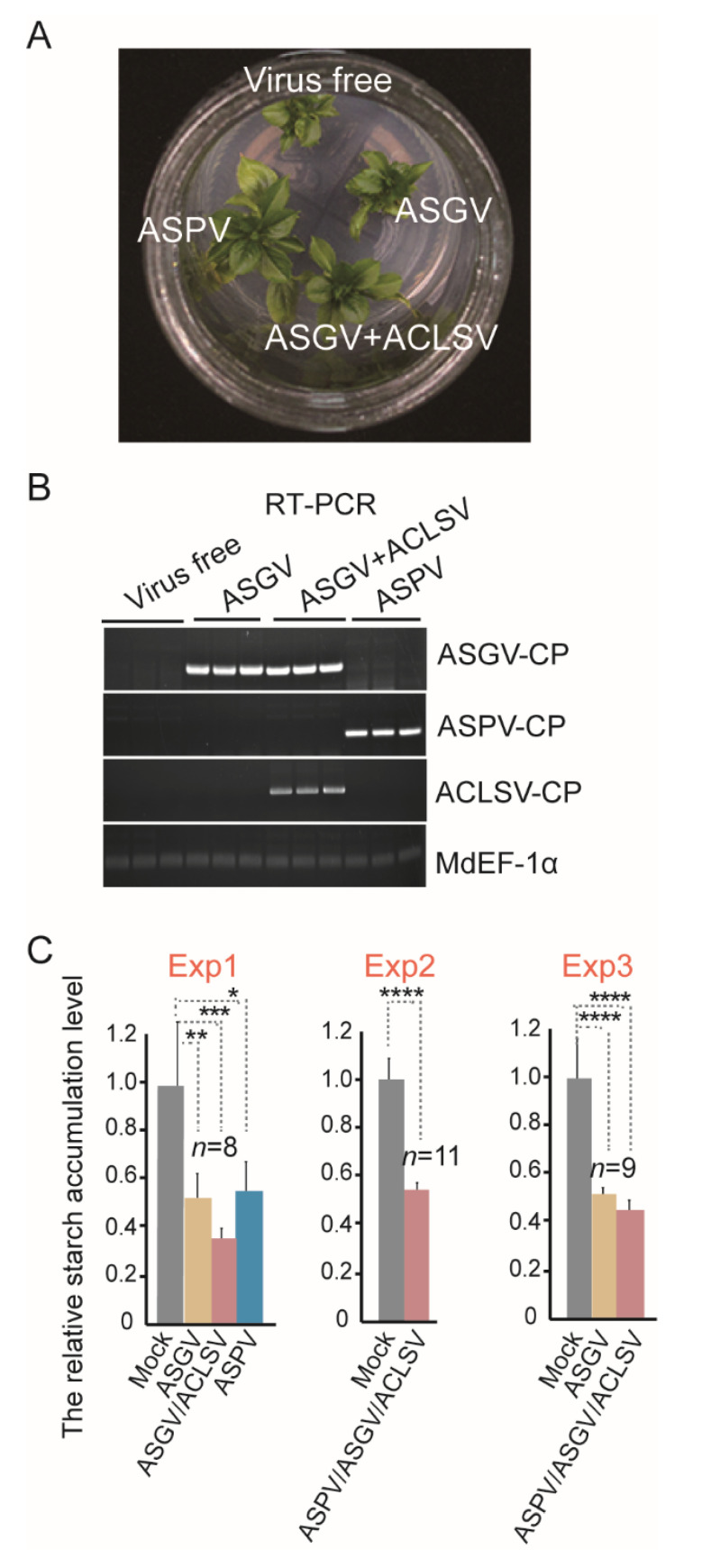
Detection of latent viruses in the propagated tissue cultures by assessing the starch content. (**A**) Shoots of apple (*Malus domestica*) propagated by tissue culture (5 weeks old). Shoots with virus infection are indicated. (**B**) RT-PCR detection of viruses in the apple shoots described in (**A**) using primer sets specific for the viral coat protein gene. Amplification of *Malus* EF-a was used as a control standard. (**C**) Relative starch accumulation in the virus-infected apple shoots described in (I). Data are presented as mean ± SD. The mock sample data value was set to 1.0. “*, **, ***”, and “****” indicate a significant difference at *p* < 0.05, 0.01, 0.001, or 0.0001, respectively (Student’s *t*-test).

**Figure 4 viruses-14-01176-f004:**
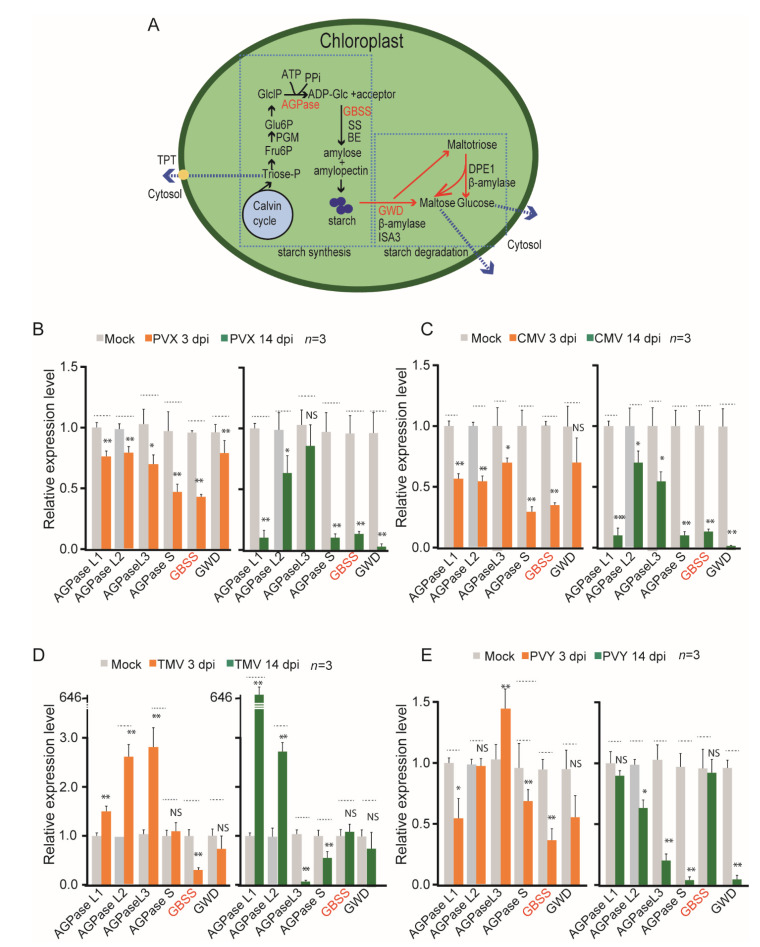
Effect of virus infection on the transcriptional expression of genes involved in starch metabolism. (**A**) A schematic diagram of the starch metabolism pathways in the chloroplast. (**B–E**) The relative transcriptional expression of genes encoding enzymes involved in starch metabolism in *N. benthamiana* plants infected with PVX, CMV, TMV or PVY. Data are presented as mean ± SD. The mock sample data value was set to 1.0. “*”, “**” and “***” indicating a significant difference at *p* < 0.05, 0.01 and 0.001 respectively. “NS” indicates no significant difference (Student’s *t*-test).

## Data Availability

Not applicable.

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
