# Peer review of "A Reduced Starch Level in Plants at Early Stages of Infection by Viruses Can Be Considered a Broad-Range Indicator of Virus Presence"

_viruses, 2022, doi:10.3390/v14061176_

Round 1

Reviewer 1 Report

The following comments need to be addressed.

  1. The authors should provide more information to explain that TMV and PVY caused a higher level of starch accumulation during the symptom development stage of virus infection. The effect of TMV and PVY on the transcriptional expression of genes, such as AGPase S, GBSS, etc. on 3 dpi and 14 dpi may give some answers.
  2. Line 96-100: “cucumber mosaic virus” (Accession No. MG025949) should add other two Accession No,since CMV has three RNAs. In addition, the resource of CMV, TMV, PVX and PVY should be re-confirmed. Based on the information of Accession No, these viruses were isolated from different regions.

Reviewer 2 Report

Comments and suggestions are in attached file. 

Reviewer 3 Report

Zhao et al. assessed the starch accumulation levels in the early stages of five plant viruses by different physiological and molecular methods in the current study. The topic of this paper sounds good, but there are some major concerns about the title, methods, results, and discussion

The manuscript is understandable. My following comments might help to improve the current MS.

Line 1 and 14, the title and the simple summary

The authors said their work was new to detecting plant viruses at the early stages of infection by assessing starch accumulation levels. I think the word new and sensitive is inaccurate because the experiments were described little; the number of viruses used in this study is too small to deduce such new results. Also, starch accumulation is not an accurate potential marker to detect the virus load as different authors reported opposite results. The brief note still needs more work to be somewhat accurate, just like you can make the TEM of the same samples used in this study to prove the reduction or increase of starch.

line 34: remove the repeated word "that"

line 36: For the keywords, plz replace the block letters with small ones except for the first-word

As minor concerns, I find the materials and methods section in need of reworking and expanding. Even when using methods reported in previous works, a brief description of the method is in order (e.g. for RNA extraction and RT PCR), and there are important details missing (e.g. sampling of the plants: which and how many leaves were collected per plant?). The section lakes data is already presented in the result and discussion. Line 109: section 2.3. must be rewritten to consider the real-time PCR data of the starch synthesis genes with their references as suddenly the authors mentioned starch genes results for the first time. Also, the treatments, collecting the samples, and the time course data are missing

line 143, results and discussion section

Figure 1A must be removed and inserted into the materials, and reconsider the shortness of its legend.

Figure 2 A the resolution is terrible. I did not see any symptoms; please replace it with another     
